# Removing the societal and legal impediments to the HIV response: An evidence-based framework for 2025 and beyond

**Anne L. Stangl**[1,2☺*], **Triantafyllos Pliakas**[3,4☺], **Jose Antonio Izazola-Licea**[5☺], **George Ayala**[6,7‡], **Tara S. Beattie**[3‡], **Laura Ferguson**[8‡], **Luisa Orza**[9‡], **Sanyukta Mathur**[10‡], **Julie Pulerwitz**[10‡], **Alexandrina Iovita**[11‡], **Victoria Bendaud**[5‡]

**1** Hera Solutions, Baltimore, MD, United States of America, **2** Department of International Health, Johns Hopkins Bloomberg School of Public Health, Baltimore, MD, United States of America, **3** Department of Public Health, Environments and Society, London School of Hygiene and Tropical Medicine, London, United Kingdom, **4** Impact Epilysis, Thessaloniki, Greece, **5** Joint United Nations Programme on HIV/AIDS (UNAIDS), Geneva, Switzerland, **6** Alameda County Public Health Department, Oakland, CA, United States of America, **7** MPact Global Action for Gay Men's Health and Rights, Oakland, CA, United States of America, **8** Institute on Inequalities in Global Health, University of Southern California, Los Angeles, CA, United States of America, **9** Frontline AIDS, Brighton, United Kingdom, **10** Population Council, Washington, DC, United States of America, **11** Global Fund to Fight AIDS, TB and Malaria, Geneva, Switzerland

☺ These authors contributed equally to this work.
‡ GA, TSB, LF, LO, SM, JP, AI and VB also contributed equally to this work.
* alstangl@gmail.com

**Data Availability Statement:** All relevant data are within the paper and its Supporting Information files.

## Abstract

Societal and legal impediments inhibit quality HIV prevention, care, treatment and support services and need to be removed. The political declaration adopted by UN member countries at the high-level meeting on HIV and AIDS in June 2021, included new societal enabler global targets for achievement by 2025 that will address this gap. Our paper describes how and why UNAIDS arrived at the societal enabler targets adopted. We conducted a scoping review and led a participatory process between January 2019 and June 2020 to develop an evidence-based framework for action, propose global societal enabler targets, and identify indicators for monitoring progress. A re-envisioned framework called the '3 S's of the HIV response: Society, Systems and Services' was defined. In the framework, societal enablers enhance the effectiveness of HIV programmes by removing impediments to service availability, access and uptake at the societal level, while service and system enablers improve efficiencies in and expand the reach of HIV services and systems. Investments in societal enabling approaches that remove legal barriers, shift harmful social and gender norms, reduce inequalities and improve institutional and community structures are needed to progressively realize four overarching societal enablers, the first three of which fall within the purview of the HIV sector: (i) societies with supportive legal environments and access to justice, (ii) gender equal societies, (iii) societies free from stigma and discrimination, and (iv) co-action across development sectors to reduce exclusion and poverty. Three top-line and 15 detailed targets were recommended for monitoring progress towards their achievement. The clear articulation of societal enablers in the re-envisioned framework should have a substantial impact on improving the effectiveness of core HIV programmes if implemented.

**Funding:** ALS and TP received funding from the Joint United Nations Programme on HIV/AIDS (UNAIDS) for the preparation of this article. JAI-L and VB are salaried employees of UNAIDS. The funders commissioned the study. JAI-L participated in the conceptualization of the study, the preparation of the manuscript and the decision to publish.

**Competing interests:** The authors have declared that no competing interests exist.

Together with the new global targets, the framework will also galvanize advocacy to scale up societal enabling approaches with proven impact on HIV outcomes.

## Introduction

In the context of HIV, an enabling environment is one free of societal, political, legal and economic impediments to availability, access and uptake of HIV services [1]. Such impediments include: stigma and discrimination, gender-based violence, punitive or harmful laws and policies, limited access to justice for key (i.e. gay men and other men who have sex with men, sex workers, transgender people and people who inject drugs) and vulnerable (i.e. women, adolescent girls, migrants, refugees and incarcerated people) populations, and gender-based, racial, economic, and educational inequalities [2, 3]. Over the past decade, emphasis has been placed on incorporating social and structural interventions, which work by altering the societal, political, legal and economic contexts that influence individual, community and societal health outcomes [4], into combination HIV prevention [5] and care and treatment strategies to improve the quality of life of people living with HIV.

In 2011, an HIV investment framework was launched to support the effectiveness and efficiency of HIV prevention, care and treatment programmes. It included a number of societal and structural interventions (described as 'critical enablers'), which, implemented alongside investments in broader programmes, such as education and poverty reduction, in different sectors (described as 'development synergies') could have a positive effect on HIV outcomes [6]. In the framework, critical enablers were divided into two groups: social enablers and programme enablers. Social enablers were defined as making environments "conducive for HIV/ AIDS responses" and programme enablers were defined as creating "demand for" and helping "improve the performance of key interventions" [6]. While the definitions were broad enough to allow for setting-specific interpretation, as these policies have been enacted, there has been a realization that greater specificity could support better decision-making about the interventions, policies, and programmes, or *societal enabling approaches*, countries should implement to increase the effectiveness of their HIV responses.

Since the publication of the HIV Investment Framework, other key guidance and initiatives have been launched that must be taken into consideration as we now refine our thinking around the enablers of the HIV response. Firstly, in 2012, UNAIDS recommended seven human rights programmes for investment to end punitive approaches to HIV: (i) reducing stigma and discrimination, (ii) increasing access to HIV-related legal services, (iii) monitoring and reforming laws, policies, and regulations, (iv) enhancing legal literacy, (v) sensitizing lawmakers and law enforcement agents, (vi) training health care providers on human rights and medical ethics related to HIV, and (vii) reducing discrimination against women in the context of HIV [7].

Secondly, in 2015, The UN launched the 17 Sustainable Development Goals, which provide a blueprint to achieve a better and more sustainable future for all by addressing the global challenges we face. The HIV response is included in Goal 3, which seeks good health and well-being, but is interconnected with a number of other goals, including Goals 1-end poverty, 2-zero hunger, 4-quality education, 5-gender equality, 8-decent work and economic growth, 10-reduced inequalities, 11-sustainable cities and communities, 16- Peace, justice and strong institutions and 17-partnership for the goals. Lastly, investments over the last four years totaling over 900 million dollars from the President's Emergency Plan

for AIDS Relief (PEPFAR) through the DREAMS programme (over 800 million) [8] and the Global Fund to Fight AIDS, Tuberculosis and Malaria (Global Fund) (123 million) [9] and others have finally made it possible for countries to support programming at sufficient scope and scale to enhance the effectiveness of HIV services by creating an enabling societal environment.

Over the past decade, significant progress has been made to develop and test interventions to address societal and legal impediments to HIV services [3, 10, 11]. This paper presents: a scoping review of the evidence on the impact of societal impediments and societal enabling approaches on HIV outcomes, a re-envisioned framework of the enablers of the HIV response, and evidence-based societal enabler targets and indicators for monitoring progress towards achieving an enabling environment for HIV services that were proposed and adopted at the UN high level meeting in June 2021.

## Methods

### Data sources and collection

The process to re-envision the enablers began with an in-house review at UNAIDS (led by JAI-L) of current understanding of how the enablers, especially the societal enablers, optimize the effectiveness of core HIV programmes (e.g. lead to increases in uptake of HIV testing, initiation of treatment, and adherence to treatment, etc.). Subsequently, and as part of a series of six technical consultations to support the 2025 target setting, a participatory multi-stakeholder technical consultation on the societal enablers took place in June 2019 [12]. Meeting participants reviewed evidence and proposed an expanded list of enablers for consideration. These included: (a) laws, policies, practices, enforcement; (b) access to justice; (c) gender equity; (d) sexual and reproductive health and rights; (e) addressing violence (prevention and response); (f) addressing HIV and key population stigma and discrimination; (g) economic justice, inequality, education, security and livelihoods (i.e. poverty, housing, work, social stability); and (h) community-led responses. While 'community-led responses' was originally proposed as a stand-alone societal enabler, we ultimately determined that it is a key service enabler, and should also be incorporated into each societal enabler, as well as in the implementation of HIV programmes, as appropriate.

Following the consultation, these eight areas were condensed further (by AS, TP and JAI-L) into overarching themes that we now consider to be the four societal enablers of the HIV response: (1) societies with supportive legal environments and access to justice, (2) gender equal societies, (3) societies free of stigma and discrimination, and (4) co-action across development sectors to reduce exclusion and poverty (Fig 1). While we recognize that other development sectors outside HIV have an impact on the HIV response, indicating the need for coordinated action at the country level, this paper focuses on the first three enablers, which fall under the purview of the HIV sector. It should be noted that the societal enablers are not mutually exclusive, and interventions are likely to focus on multiple enablers. Success in one societal enabler (e.g. supportive legal environments) is very likely to influence another (e.g. reduced HIV stigma and discrimination).

A scoping review was then performed on research published in English up to 16 June 2020. This type of review was chosen due to the diversity of evidence across the broad range of societal enablers that we were attempting to clarify [13]. The purpose of the review was to identify the best available evidence regarding the impact of societal impediments (e.g. criminalization, violence, stigma and discrimination, etc.) and societal enabling interventions (de-criminalization; violence reduction, etc.) on HIV outcomes to inform the re-envisioned framework. We searched available published literature across three databases: Pubmed, Scopus and Web of

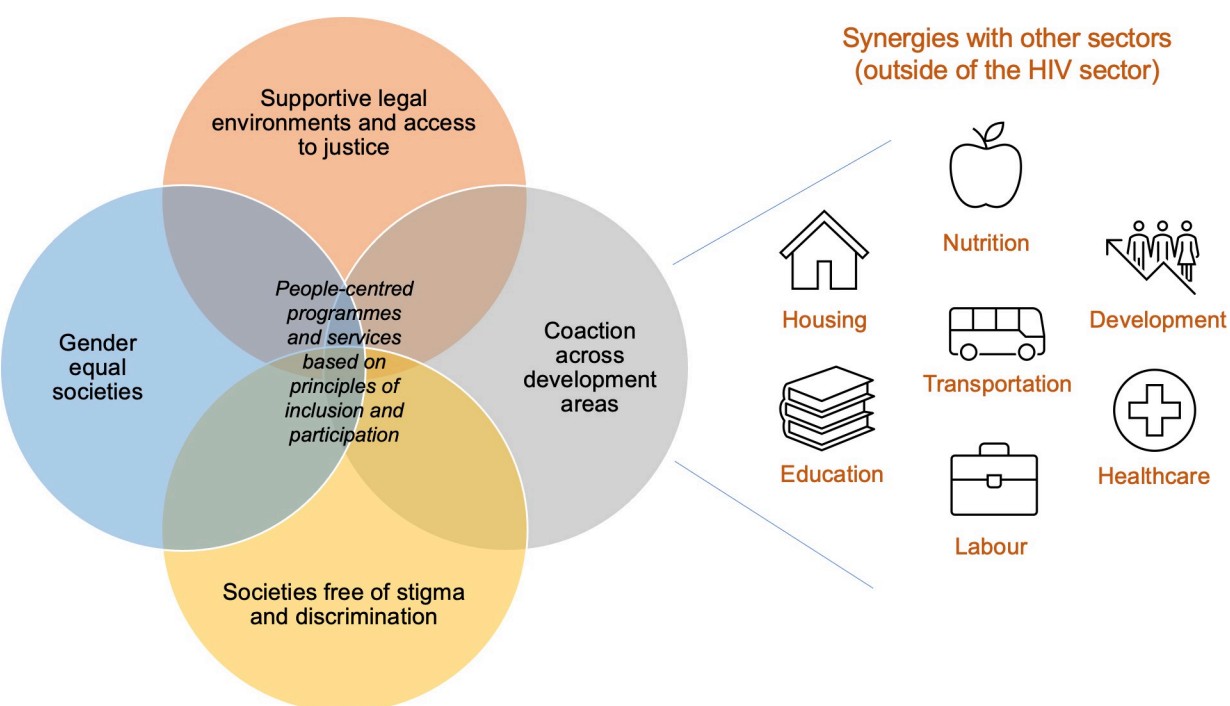

**Fig 1. The societal enablers of the HIV response.**

Science. The Population, Intervention, Comparison and Outcome (PICO) framework was used to develop the search strategy. We developed three blocks of search terms to capture the populations of interest, the societal impediments and/or societal enabling approaches and HIV outcomes. Specific search terms used are available in S1 Table. We included all study designs across all countries and population groups. For this paper, we include only peer-reviewed studies that explicitly examined the relationship between a societal enabler or impediment and an HIV outcome/s and demonstrated a significant impact using quantitative measures.

One author (TP) screened the title and abstract for all records and a second author (AS) examined a random selection of records. Expert advice from the Technical Expert Group on Social Enablers and HIV and UN co-sponsors added additional articles not captured in the literature search. We extracted information from articles related to the study author, the year of publication, the country, the study design, the study population and sample, the social impediment studied/addressed, the intervention description, duration and socio-ecological level of the intervention where appropriate, the HIV outcome/s, and impact estimates of the societal impediment or societal enabling approach on HIV outcomes. We examined HIV outcomes including HIV prevalence, HIV incidence, HIV testing, ART adherence, AIDS-related mortality, linkage to HIV care and viral suppression.

We limited our search strategy to the three enablers that fall within the HIV sector (S1 Fig). Development coaction areas (i.e. education, poverty reduction and economic development) that influence HIV outcomes have already been clearly described in the Sustainable Development Goals (SDGs) and existing evidence-based targets are available [14]. Evidence from 16 studies on the impact of key development co-action areas on HIV outcomes was recommended by technical experts and UN co-sponsors and is summarized in S2 Table.

## Results

A total of 30 studies met the inclusion criteria and are described in Table 1. Most studies (60%; N = 18/30) examined societal impediments to the HIV response, rather than societal enabling approaches. We review the evidence by societal enabler.

### Societies with supportive legal environments and access to justice

All six studies reviewed on the legal environment assessed the impact of a societal enabling approach on an HIV outcome/s. The evidence reviewed highlighted the positive impact of decriminalisation of occupations and behaviors that heighten an individual's risk of being exposed to HIV, including sex work, drug use and same-sex behavior. For example, decriminalising sex work could avert 33–46% of HIV infections among female sex workers in the next decade across all settings [19]. Similarly, modelling data from Mexico suggest that implementing law reform would reduce incarceration in people who inject drugs by 80% from 2018 onward, averting 9% of new HIV infections between 2018 and 2030, with 21% averted if people who inject drugs were referred to opioid agonist treatment instead of being incarcerated [16].

A recent systematic review and meta-analysis of pooled data on HIV testing and engagement with the HIV treatment cascade among African men who have sex with men revealed that levels of testing ever, in the past 12 months and status awareness were significantly lower in countries with the most severe anti-lesbian, gay, bisexual and transgender legislation, compared to countries with the least severe legislation [20]. Likewise, the Same-Sex Marriage Prohibition Act passed in Nigeria in 2014 significantly increased fear of accessing healthcare services among men who have sex with men [18]. Supportive legislation, however, such as the gender identity law passed in Argentina in 2012, which among other things made it easier for people to legally change their gender identity, can reduce stigma and discrimination towards key populations, increase HIV testing and improve quality of life [15]. Similarly, legislation reducing the age of consent for accessing HIV testing to less than 16 has been linked with 11.0 percentage points higher coverage of HIV testing among youth [17]. We did not identify any quantitative evidence of the impact of access to justice interventions on HIV outcomes.

### Gender equal societies

Seven studies, including two systematic reviews, examined the impact of gender equality-related societal impediments on HIV outcomes, including experience of any physical or sexual violence, violence from non-partners, intimate partner violence (IPV), and inequitable gender norms. Experience of any violence has been linked to reduced condom use with clients among female sex workers in India [21]. Likewise, female sex workers who experience violence from non-partners (clients, police, etc.) have an increased risk for HIV [aOR (95%CI): 1.59 (1.18, 2,15)) in India [22]. IPV has also been linked with a higher risk of acquiring HIV among women in the U.S., with 11.8% of HIV infections among women attributable to IPV in the past year [28]. This finding is supported by a systematic review of the association of IPV with engagement in care, which found significant associations with lower odds of current ART use [OR (95% CI) 0.79: (0.64–0.97)], ART adherence [OR (95% CI): 0.48 0.30–0.75)] and viral suppression [(OR (95% CI): 0.64 (0.46–0.90)] [23]. In addition, a systematic review and meta-synthesis of 28 studies from 16 countries found a moderate statistically significant association between IPV and HIV infection among women, including physical violence [Pooled RR (95% CI): 1.22 (1.01,1.46)] and any type of violence (i.e. physical, sexual, psychological) [Pooled RR (95% CI): 1.28 (1.00, 1.64) [25].

Modelling data suggest that the elimination of sexual violence alone could avert 17% of HIV infections in Kenya and 20% in Canada, through its immediate and continued effect on

**Table 1. Study and intervention characteristics, HIV outcomes assessed, and study findings by societal enabler from 30 studies.**

| 1st Author, publication date, country, study design[A] | Study Population[B] | Sample | Intervention/Policy Description, duration | Socio-ecological Levels | HIV Outcomes | Results (Positive, Negative, No effect; Details) |
|---|---|---|---|---|---|---|
| *Supportive legal environments and access to justice (n = 6)* | | | | | | |
| Aristegui 2014, Argentina, (QS) [15] | Transgender people | Two focus groups with 20 transgender women | Gender Identity law adopted in 2012 | Public Policy | HIV testing; quality of life; stigma and discrimination | *Positive* |
| | | | | | | Better and earlier access to health services among transgender people, including HIV testing and treatment. |
| | | | | | | Reduction in stigma and discrimination in health-care settings: only three out of 10 study participants reported discrimination based on their gender identity after the enactment of the law (compared to eight out of 10 before it). |
| | | | | | | Quality of life of transgender people, increasing their access to education, work and health services. |
| Borquez, 2018, Mexico, MS [16] | PWID | 733 | Drug law reform, which de-penalised the possession of small amounts of drugs and instituted drug treatment instead of incarceration | Individual | HIV infections | *Positive* |
| | | | | | | Modelling estimated the limited reform implementation averted 2% (95% CI 0·2–3·0) of new HIV infections |
| | | | Evaluating impacts between 2012 and 2017 | | | If implementation reduced incarceration in people who inject drugs by 80% from 2018 onward, 9% (95% CI 4–16) of new HIV infections between 2018 and 2030 could be averted, with 21% (10–33) averted if people who inject drugs were referred to opioid agonist treatment instead of being incarcerated. |
| McKinnon, 2019, sub-Saharan Africa, PS-M [17] | Adolescents aged 15–18 | 62,628 adolescents, of which 39 339 were females and 23 289 were males, across 15 countries | Evaluating impact of legal age of consent on coverage of HIV testing among adolescents between 2011–2016 | Public Policy | HIV testing | *Positive* |
| | | | | | | Legal age of consent below 16 years was associated with an 11.0 percentage points higher coverage of HIV testing (95% CI: 7.2 to 14.8 corresponding to a rate ratio of 1.74 (1.35–2.13). |
| | | | | | | HIV testing rate had a stronger association with lower age of consent among females than males. The testing rates differences were 14.0 percentage points (8.6–19.4) for females and 6.9 percentage points (1.6–12.2) for males (P-value for homogeneity = 0.07). |
| Schwartz, 2015, Nigeria, B/A [18] | MSM | 707 | TRUST is a prospective implementation research cohort study. | Individual | Fear of accessing healthcare | *Negative* |
| | | | | | | MSM were more likely to fear accessing healthcare following the enactment of legislation to further criminalising same-sex practices |
| | | | Before and after implementation of the Same-Sex Marriage Prohibition Act Mar 2013 –Aug 2014 | | | Fear of seeking health care |
| | | | | | | (aIRR: 2.92, 95% CI 1.46–5.84) |
| | | | | | | No safe spaces to be with other MSM |
| | | | | | | (aIRR: 3.26, 95% CI 1.94–5.48) |
| Shannon, 2015, SR and MS [19] | FSW | 87 studies designed a priori to examine one or more structural determinants of HIV, HIV and sexually transmitted infection (STI), or condom use | Varied across studies | Varied across studies | HIV infections | *Positive* |
| | | | | | | Decriminalisation of sex work would have the greatest effect on the course of HIV epidemics across all settings, averting 33–46% of HIV infections in the next decade. |
| Stannah, 2019, Africa, SR-MA [20] | MSM | 44,993 MSM from 75 independent studies | Anti-LGBT Legislation using four anti-LGBT legislation variables: repressive legislation, lack of protective legislation, lack of progressive legislation, and a penalties variable (score 0–14 with higher scores reflecting less progressive legislation). | Varied across studies | Ever tested | *Negative* |
| | | | | | | Decreased by 2% (95% CI 1–4%) for each point increase on the global anti-LGBT legislation index |
| *Gender equal societies (n = 9)* | | | | | | |

*(Continued)*

**Table 1.** (Continued)

| 1st Author, publication date, country, study design[A] | Study Population[B] | Sample | Intervention/Policy Description, duration | Socio-ecological Levels | HIV Outcomes | Results (Positive, Negative, No effect; Details) |
|---|---|---|---|---|---|---|
| Beattie, 2010, India, B/A [21] | FSW | 3,852 | A multi-layered strategy involving policy makers, secondary and primary stakeholders, to stem and address violence against the sex worker community as part of a wider HIV intervention program, examine the impact of these violence intervention efforts on levels of violence against FSWs, and examine associations between violence and condom use, HIV/STI rates and exposure to the HIV prevention program components. Baseline integrated behavioural and biological assessments were conducted 12–16 months after program initiation, and follow-up surveys completed 33–37 months later. | Individual, Community, Policy | HIV infections, condom use | *Negative* <br><br> Violence in the past year was not significantly associated with HIV infection but strongly associated with reduced condom use with clients <br><br> *HIV-1 infection* <br> OR: 1.10 (0.80–1.49), p = 0.60 <br> aOR: 0.96 (0.70–1.32), p = 0.80 <br><br> *Condom use last sex act occasional clients* <br> OR: 0.75 (0.53,1.07), p = 0.10 <br> aOR: 0.58 (0.40–0.85), p = 0.005 <br><br> *Condom use last sex act repeat clients* <br> OR: 0.48 (0.35–0.67), p<0.001 <br> aOR: 0.49 (0.35–0.70), p<0.001 <br><br> *Condom use last sex act regular partner clients* <br> OR: 1.14 (0.81–1.61), p = 0.50 <br> aOR: 0.86 (0.54–1.37), p = 0.50 <br><br> *Condom use last anal sex* <br> OR: 0.69 (0.40–1.19), p = 0.20 <br> aOR: 0.69 (0.40–1.21), p = 0.20 |
| Beattie, 2015, India, O/RXS [22] | FSW | 5,792 FSWs participated in the Integrated Bio-Behavioral Assessments and 15,813 FSWs participated in the polling booth surveys | Avahan programme | Community | HIV prevalence | *Negative* <br><br> Experience of non-partner violence (being raped in the past year and/or beaten in the past six months) was significantly associated with HIV prevalence <br><br> aOR: 1.59 (1.18, 2.15), p = 0.002 |
| Hatcher, 2015, SR-MA [23] | Women living with HIV | 3,365 from two countries (Haiti and USA) in 13 O/XS studies | No intervention | Not applicable | Treatment adherence Viral suppression | *Negative* <br><br> Intimate partner violence significantly associated with lower ART use, poorer self-reported ART adherence and lower odds of viral load suppression <br><br> *ART use* <br> OR = 0.79 (0.64–0.97) <br><br> *ART adherence* <br> OR = 0.48 (0.30–0.75) <br><br> *Viral suppression* <br> OR = 0.64 (0.46–0.90) |
| Kyegombe, 2014, Uganda, CRT [24] | General population | 1,583 men and women at baseline and 2,532 at follow-up were interviewed | SASA! community mobilization intervention focused upon shifting harmful social norms, addressing the power imbalances between women and men, HIV-related risk and inequitable relationships; selected community members actively discussed and engaged on issues of gender inequality, violence and HIV (community members, healthcare workers, police, govt leaders). The study took place between 2007 and 2012. | Community | HIV testing, condom use | *Positive* <br><br> Increase in HIV testing and condom use among men <br><br> *HIV testing in past year* <br> Women <br> RR: 1.01 (0.92–1.12), aRR: 1.02 (0.89–1.15) <br> Men <br> RR: 1.54 (1.15, 2.05), aRR: 1.50 (1.13–2.00) <br><br> *Condom use in past year* <br> Women <br> RR: 1.15 (0.79–1.69), aRR: 1.22 (0.90–1.66) <br> Men <br> RR 1.52 (1.04–2.20), aRR: 1.54 (0.96–2.47) <br><br> *Condom use at last intercourse* <br> Women <br> RR: 1.37 (0.59–3.20), aRR: 1.58 (0.86–2.89) <br> Men <br> RR: 1.91 (1.13–3.23), aRR: 2.03 (1.22–3.39) |

*(Continued)*

**Table 1.** (Continued)

| 1st Author, publication date, country, study design[A] | Study Population[B] | Sample | Intervention/Policy Description, duration | Socio-ecological Levels | HIV Outcomes | Results (Positive, Negative, No effect; Details) |
|---|---|---|---|---|---|---|
| Li, 2014, SR-MA [25] | General population | 331,468 women from 16 countries in 28 studies (19 O/XS, 5 O/RXS and 4 CCS) | Varied across studies | Varied across studies | HIV infection | *Positive* |
| | | | | | | Physical intimate partner violence and any type of intimate partner violence were significantly associated with HIV infection in cohort and cross-sectional studies |
| | | | | | | Cohort studies |
| | | | | | | *Physical intimate partner violence* |
| | | | | | | Pooled RR: 1.22 (1.01–1.46) |
| | | | | | | *Any type of intimate partner violence* |
| | | | | | | Pooled RR: 1.28 (1.00–1.64) |
| | | | | | | Cross-sectional studies |
| | | | | | | *Physical intimate partner violence* |
| | | | | | | Pooled RR: 1.44 (1.10–1.87) |
| | | | | | | *Combination of physical and sexual intimate partner violence* |
| | | | | | | Pooled RR: 2.00 (1.24–3.22) |
| | | | | | | *Any type of intimate partner violence* |
| | | | | | | Pooled RR: 1.41 (1.16–1.73) |
| Mohlala, 2011, South Africa, RCT [26] | Pregnant women (and partners) | 304 | Male participation in antenatal care and uptake of couple voluntary counselling and testing for HIV. Partners received invitation for voluntary counselling and testing (VCT) or pregnancy information sessions (PIS). Two study/couple visits took place, 1 and 12 weeks after randomization. | Individual, Interpersonal | HIV infection | *Positive* |
| | | | | | | More partners with HIV testing |
| | | | | | | HIV infection status (comparing infected vs not infected) |
| | | | | | | OR: 1.53 (1.16–2.03), p = 0.003 |
| | | | | | | aOR: 1.50 (1.11–2.02), p = 0.007 |
| Pulerwitz, 2019, South Africa, O/XS [27] | Men and women aged 18–49 | 970 women and 979 men | No intervention | Not applicable | HIV testing and ART treatment | *Positive* |
| | | | | | | Endorsement of inequitable gender norms was associated with more testing in women but not in men. Endorsement of inequitable gender norms among people living with HIV was associated with less current treatment use for both women and men |
| | | | | | | *HIV testing* |
| | | | | | | Women, aOR: 2.47 (1.46–4.18), p < 0.01 |
| | | | | | | Men, aOR: 1.38 (0.95–2.01), p > 0.05 |
| | | | | | | *Current ART* |
| | | | | | | Women, aOR: AOR 0.15 (0.04–0.53), p < 0.01 (full GEMS) |
| | | | | | | Men, aOR: 0.57 (0.08–3.82), p>0.05 (full GEMS) |
| | | | | | | Men, aOR: 0.28 (0.08, 0.93), p<0.05 (norms around men as the decision maker in a couple) |
| Sareen, 2009, USA, O/XS [28] | Women in general population | 13,842 | No intervention | Not applicable | HIV infections | *Negative* |
| | | | | | | Intimate partner violence was significantly associated with HIV infection |
| | | | | | | OR = 5.79 (2.10–15.97), p<0.01 |
| | | | | | | aOR = 3.44 (1.28–9.22), p<0.05 |
| Shannon, 2015, SR and MS [19] | FSW | 87 studies designed a priori to examine one or more structural determinants of HIV, HIV and sexually transmitted infection (STI), or condom use | Varied across studies | Varied across studies | HIV infections HIV condom use | *Positive* |
| | | | | | | This modelling suggested that elimination of sexual violence alone could avert 17% of HIV infections in Kenya (95% uncertainty interval [UI] 1–31) and 20% in Canada (95% UI 3–39) through its immediate and sustained effect on non-condom use) among FSWs and their clients in the next decade |
| *Societies free of stigma and discrimination (n = 15)* | | | | | | |
| Boyer, 2011, Cameroon, O/XS [29] | PLHIV | 2,117 | No intervention | Not applicable | Treatment adherence | *Negative* |
| | | | | | | aOR:f 1.74, 95% CI 1.14–2.65 |
| Chimoyi, 2015, South Africa, O/XS [30] | Commuters from general population | 1,146 | No intervention | Not applicable | HIV testing | *Negative* |
| | | | | | | Stigma and discrimination reduced the likelihood of testing |
| | | | | | | aOR: 0.40 (0.31–0.62) |

*(Continued)*

**Table 1.** (Continued)

| 1st Author, publication date, country, study designA | Study PopulationB | Sample | Intervention/Policy Description, duration | Socio-ecological Levels | HIV Outcomes | Results (Positive, Negative, No effect; Details) |
|---|---|---|---|---|---|---|
| Christopoulos, 2019, USA, O/RXS [31] | PLHIV | 6,448 | No intervention | Not applicable | Viremia | *Positive* <br><br> Mean stigma score was associated with concurrent viremia <br><br> aOR: 1.13 (1.02–1.25) |
| Dalrymple, 2019, Scotland, Wales, Northern Ireland and Republic of Ireland, O/XS [32] | MSM | 2,436 | No intervention | Not applicable | HIV testing | *Negative* <br><br> Higher personalised stigma score was associated with reduced odds for HIV testing <br><br> aOR: 0.97 (0.94–1.00) |
| Gesesew, 2017, SR-MA [33] | PLHIV | 3,788 persons from 10 studies | Varied across studies | Varied across studies | Linkage to HIV care | *Negative* <br><br> PLHIV perceiving high levels HIV-related stigma were two times more likely to present late for HIV care compared to PLHIV experiencing low levels of HIV-related stigma <br><br> (Pooled OR: 2.4, 95% CI 1.6–3.6, $I^2$ = 79%) |
| Golub and Gamarel, 2013, USA, O/XS [34] | LGBTQ | 305 | No intervention | Not applicable | HIV testing | *Negative* <br><br> MSM and transgender women experiencing anticipated stigma were 46% less likely to test for HIV in the past six months <br><br> (aOR: 0.54, 95% CI 0.40–0.73) |
| Hargreaves, 2020, Zambia and South Africa, CRT [35] | PLHIV | 3,963 | 4-year HIV combination prevention intervention trial <br><br> Did not include stigma reduction strategies | Community; Individual | Viral suppression among people living with HIV taking ART | *Negative* <br><br> PLHIV experiencing internalized stigma were less likely to be virally suppressed <br><br> aRR: 0.94, 95% CI 0.89–0.98 <br><br> *No effect* <br><br> Experienced or perceived stigma among PLHIV was not associated with viral suppression <br><br> Experienced stigma in health service settings <br><br> aRR: 0.99, 95% CI 0.93–1.06 <br><br> Experienced stigma in the community <br><br> aRR: 0.98, 95% CI 0.94–1.02 <br><br> Perceived stigma in health service settings <br><br> aRR: 1.05, 95% CI 0.96–1.15 <br><br> Perceived stigma in the community <br><br> aRR: 1.01, 95% CI 0.94–1.10 |
| Langebeek, 2014, SR-MA [36] | Varied across studies | 207 studies | Varied across studies | Varied across studies | ART adherence | *Negative* <br><br> In 47 of 207 studies, HIV stigma associated with ART adherence <br><br> Standardized mean difference with standard error: -0.282 (0.038). |
| Lipira, 2019, USA, O/XS [37] | African American women living with HIV | 100 | Baseline results from a multisite randomized controlled trial testing the effectiveness of a behavioral intervention to reduce HIV-related stigma among African American women living with HIV | Individual | Viral suppression | *Negative* <br><br> Higher levels of HIV-related stigma were associated with lower odds of being virally suppressed <br><br> aOR = 0.93, 95% CI = 0.89–0.98 |
| Kemp, 2019, USA, RCT [38] | African American women living with HIV | 234 | A multi-site randomized controlled trial testing the effectiveness of a behavioral intervention (a workshop that met for 4–5 h during 2 consecutive weekday afternoons) to reduce HIV stigma among African American women living with HIV | Individual | Viral load | *Negative* <br><br> HIV stigma (enacted and internalized stigma) was significantly associated with subsequent viral load (adjusted b = 0.24, P = 0.005). <br><br> Both between-subject (adjusted b = 0.74, P<0.001) and within-subject (adjusted b = 0.34, P = 0.005) differences in enacted stigma were associated with viral load. |
| Katz, 2013, SR-MS [39] | PLHIV | 26,715 persons from 32 countries in 75 studies (34 qualitative, 41 quantitative) | Varied across studies | Varied across studies | Treatment adherence | *Negative* <br><br> 24 of 33 cross-sectional studies (71%) reported a positive finding between HIV stigma and ART non-adherence <br><br> *No effect* <br><br> 6 of 7 longitudinal studies (86%) reported a null finding between HIV stigma and ART non-adherence |

*(Continued)*

**Table 1.** (Continued)

| 1st Author, publication date, country, study design[A] | Study Population[B] | Sample | Intervention/Policy Description, duration | Socio-ecological Levels | HIV Outcomes | Results (Positive, Negative, No effect; Details) |
|---|---|---|---|---|---|---|
| Peitzmeier, 2015, The Gambia, O-XS [40] | PLHIV | 317 | No intervention | Not applicable | Linkage to care and non-use ART | *Negative* |
| | | | | | | Enacted stigma in health care settings was significantly associated with avoiding or delaying seeking care. Enacted stigma in the household or community and internal stigma were marginally associated |
| | | | | | | Enacted stigma in health care setting |
| | | | | | | aOR = 3.03 (1.24–7.89) |
| | | | | | | Enacted stigma in the household or community |
| | | | | | | aOR = 1.21 (0.98–1.49) |
| | | | | | | Internal stigma |
| | | | | | | aOR = 1.47 (0.96–2.22) |
| | | | | | | Enacted stigma in health care settings was significantly associated with non-use of antiretroviral therapy, whereas internal stigma and enacted stigma in the household or community were not. |
| | | | | | | Enacted stigma in the household or community |
| | | | | | | aOR = 0.52 (0.31–0.88) |
| Sabapathy, 2017, Zambia and South Africa, CCS [41] | PLHIV | 705 | Uptake of universal treatment, specifically timely linkage-to-care and initiation of treatment following door-to-door universal testing, during the first year of the PopART universal test and treat intervention. | Community; Individual | Linkage to care and treatment initiation | *Negative* |
| | | | | | | PLHIV who have felt ashamed of their HIV status are more likely of late presentation for HIV care and late treatment initiation |
| | | | | | | (aOR: 1.82, 95% CI 1.10–3.03 if they agree to the statement |
| | | | | | | aOR: 1.71, 95% CI 1.05–2.79 if they strongly agree to the statement) |
| Weiser, 2006, Botswana, O/XS [42] | Community members | 1,268 | No intervention | Not applicable | HIV testing | *Negative* |
| | | | | | | Individuals with stigmatizing attitudes toward people living with HIV and AIDS were less likely to have been tested for HIV |
| | | | | | | aOR = 0.7 (0.5–0.9) |
| Zulliger, 2015, Dominican Republic, O/XS [43] | FSW living with HIV | 268 | No intervention | Not applicable | ART interruption | *Positive* |
| | | | | | | The odds of ART interruption were higher among women who experienced FSW-related discrimination and had higher internalized stigma |
| | | | | | | *FSW-related discrimination* |
| | | | | | | aOR = 3.24 (1.28–8.20) |
| | | | | | | *Internalized stigma* |
| | | | | | | aOR = 1.09 (1.02–1.16) |

[A] Study design abbreviations: B/A: Before/after study; CRT: Cluster randomised trial; CCS: Case-control study; MM = mixed methods; MS: Modelling study; O/XS = observational cross-section; O/RXS = observational repeated cross-sections; PR = policy review; PMD = program monitoring data; RCT: Randomised controlled trial; QP = qualitative post-test only; SR: Systematic review; SR-MA: Systematic review with meta-analysis; SR-MS: Systematic review with meta-synthesis

[B] HCW = healthcare workers; LGBTQ = lesbian, gay, bisexual, transgender, and questioning; PLHIV = people living with HIV; PWID = people who inject drugs; SW = sex workers; aRR: adjusted relative risk; aOR: adjusted odds ratio; CI: Confidence intervals; OR: Odds ratio; $I^2$: testing the statistical heterogeneity among the studies; IRR: Incidence rate ratio.

non-condom use among female sex workers and their clients in the next decade [19]. No studies were identified that examined the association of IPV or gender-based violence, or the impact of interventions to reduce such violence, with HIV outcomes among other key populations, such as gay men and other men who have sex with men and transgender people. A study in South Africa that examined the influence of inequitable gender norms on HIV service use behaviours found that both women and men living with HIV who endorsed inequitable gender norms were less likely to be currently taking antiretrovirals, (i.e., women who endorsed

norms accepting men's control over and violence towards women; men as the main / sole deci-sion-maker in a couple; and men as reluctant to seek care/help during illness; and men who endorsed norms around men as the main/sole decision maker in a couple). This study also found that receiving an HIV test in the past year was significantly associated with endorsement of inequitable gender norms (among women only, and especially for norms suggesting women have the primary/exclusive responsibility as family caretaker). While unexpected, additional analyses conducted by the study authors suggested that the association was likely due to the greater likelihood of testing after having children/during pregnancy, as HIV testing is rou-tinely offered at antenatal services in South Africa, and as women with children were more likely to endorse those primary caretaker norms [27].

Two studies assessed the impact of social enabling approaches to improve gender equality on HIV outcomes. Community mobilization interventions to reduce IPV led to increased HIV testing and condom use among heterosexual men in Uganda [24]. Likewise, heterosexual cou-ples HIV counselling and testing in South Africa led to more partners testing for HIV and learning their HIV status [26].

## Societies free of stigma and discrimination

All 15 studies included examined the impact of different domains of stigma and discrimination on HIV outcomes, rather than the impact of a societal enabling approach. Only two studies examined the link between key population specific stigma and discrimination and HIV out-comes, one with female sex workers [43] and one with gay men and other men who have sex with men [32]. The evidence reviewed from 12 studies and 3 systematic reviews found a nega-tive impact of HIV and key population stigma and discrimination on linkage to HIV care [33, 41], HIV testing among the general population [30, 42], HIV testing among the lesbian, gay, bisexual, and transgender community [32, 34], viral suppression [31, 35, 37, 38, 44], treatment adherence [29, 36, 39] and treatment initiation [41]. Experienced stigma in the healthcare set-ting was also linked with avoiding or delaying care seeking for HIV [40].

Specifically, anticipated stigma if a test result is positive impedes HIV testing [34] and inter-nalized stigma, where people living with HIV, or people belonging to a key population group, apply negative feelings to themselves, has been linked with refusal to accept ART among newly diagnosed people living with HIV [41]. Similarly, people living with HIV who perceived high HIV stigma were twice as likely to delay enrolment in HIV care than those who perceived low HIV stigma [33] and men who have sex with men who reported stigma related to being gay had reduced odds of HIV testing [32]. Internalized stigma also impedes ART adherence among people living with HIV and key populations by compromising social support and adap-tive coping [39, 43], and has been linked to poorer viral suppression among people living with HIV who are taking antiretroviral therapy (ART) [35, 37, 38, 44]. Among female sex workers living with HIV, experienced discrimination related to being a sex worker was associated with higher odds of ART interruption [43].

## The Society-, System- and Service-enablers of the response to HIV: The 3 S's

We re-examined the 2011 HIV Investment framework with the four societal enablers in mind and found that the critical enablers could be better organized based on what they enable: HIV services, HIV systems or the social environment in which the HIV response is being imple-mented. Thus, in the new framework, enablers are differentiated based on: society, systems and services (abbreviated as the 3 S's) (Fig 2).

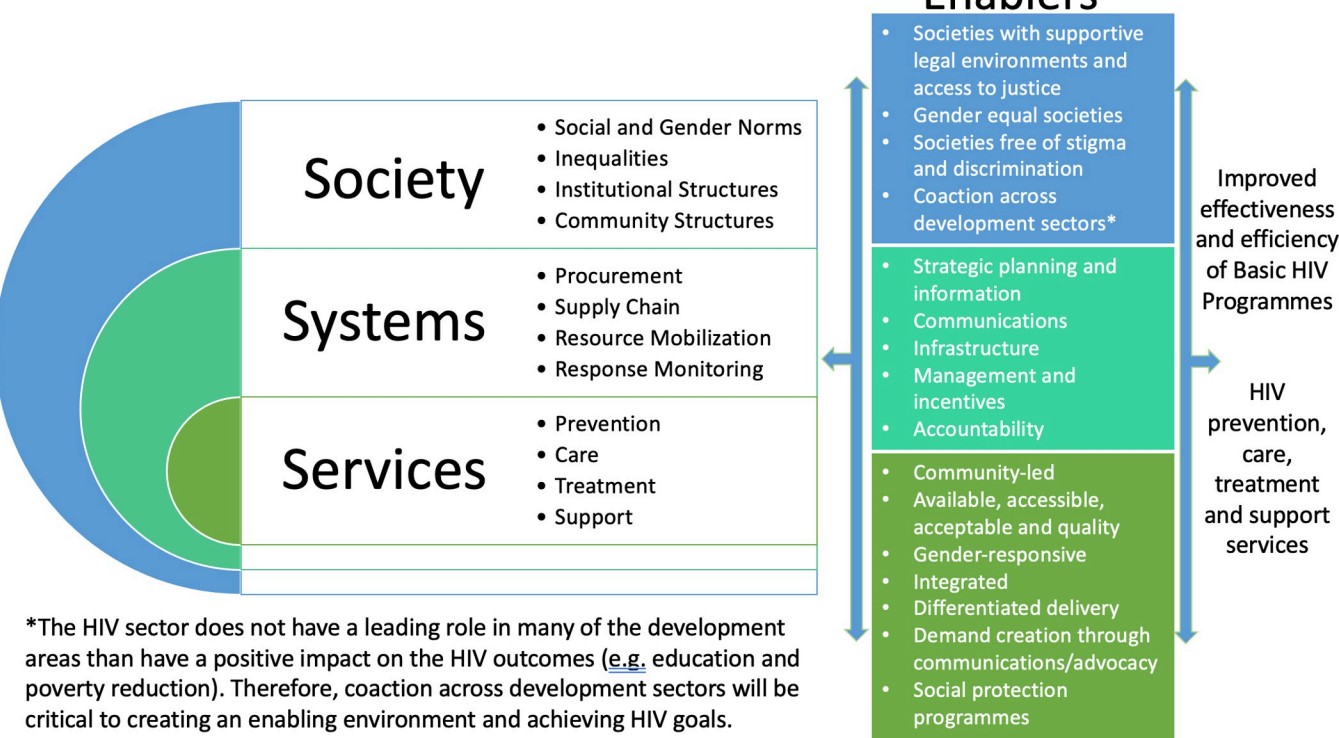

**Fig 2. The 3 S's of the HIV response: A new framework for conceptualising enablers of HIV services and systems and the social environment in which they operate.**

Service enablers include interventions to increase the availability, accessibility, acceptability and quality of HIV prevention, care, treatment and support services [45]. Such enablers also ensure that HIV services are non-discriminatory, gender-responsive, integrated where needed and differentiated–a person-centred approach that "simplifies and adapts HIV services across the cascade, in ways that both serve the needs of people living with HIV better and reduce unnecessary burdens on the health system" [46]. Service enablers also take into account the principles of participation and inclusion, including service provision that is led by or involves the communities of people living with and affected by HIV [47], tapping into community innovations [48]. In addition, service enablers include programmes to create demand for HIV services through communications and advocacy and social protection programmes, such as housing, nutrition, and public transportation, that enhance the effectiveness of HIV service uptake among marginalized communities. System enablers, health or otherwise, include broader strategies, approaches or functions to improve efficiencies in procurement and supply chains, resource mobilization and response monitoring. Such enablers include strategic planning and information, communications, infrastructure, management, and incentives and accountability.

The social environment can greatly influence how well countries are able to implement HIV systems and services [49]. Enabling approaches at the societal level are interventions, programmes or policies that improve the response to HIV. National governments and development partners should invest substantially in societal enabling approaches to achieve the four overarching societal enablers, heeding the call for co-action with the broader social development programmes. This call includes the need to reduce poverty and increase nutrition,

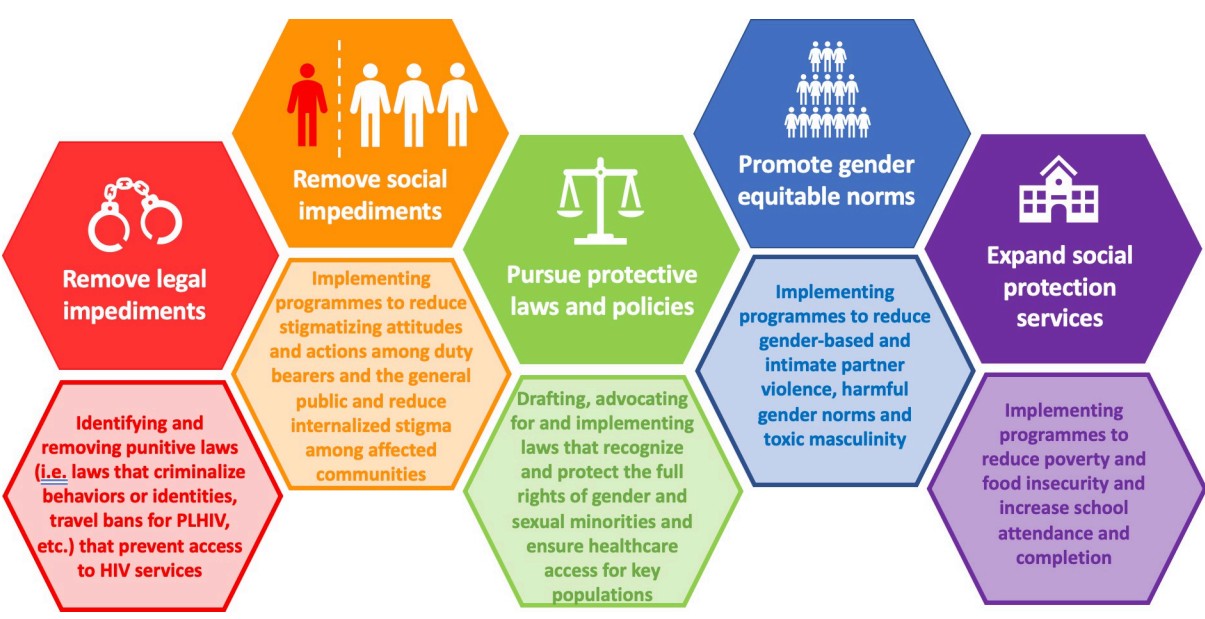

**Fig 3. A societal enabling continuum to increase effectiveness HIV services.**

education, and access to housing, transportation and decent work with evidence-based strategies identified and funded by appropriate development agencies.

Achieving an enabling societal environment is a process, reflected as a continuum in Fig 3. Ideally, countries will focus first on removing legal and societal impediments to HIV services, and then turn towards expanding legal protections for marginalized populations, promoting gender equitable norms, and expanding social protection through policies and programming. However, we recognize that countries are at different stages and determining where to target investments in societal enabling approaches will vary by context.

## Proposed targets for monitoring progress on societal enablers or lifting impediments

Based on the evidence reviewed and input from technical experts, we proposed a set of targets to inform HIV response planning to create an enabling environment for HIV programmes. Three top-line and 15 detailed targets were selected in addition to expressing the need for simultaneous action across the development sectors to achieve the SDGs linked with HIV outcomes (S3 Table). The monitoring framework to assess progress towards these targets includes 15 indicators, seven of which have been included in Global AIDS Monitoring (GAM) previously, five of which have been added to the 2022 GAM guidance, one of which is being finalized, and two of which are being piloted with the expectation of adding them to the 2023 GAM guidance. Baseline data are available for several countries for nine indicators, a few countries for four indicators, and no countries for two indicators (Table 2). While data were not available for all proposed targets, we ultimately proposed three, aggressive top-line targets given the urgent need to achieve enabling social environments to achieve the 2030 HIV goals, including: (1) Less than 10% of countries have legal environments that impede HIV services; (2) Less than 10% of women, girls and key populations experience gender inequality and violence; and (3) Less than 10% of people living with HIV and key populations experience stigma and discrimination.

It should be noted that based on available GAM data, some countries are closer to achieving an enabling societal environment than others (Table 2). For example, while a median of 56.6% of the general population report discriminatory attitudes towards people living with HIV, discriminatory attitudes range from 12.7% to 75.7% across countries (S2 and S3 Figs). We recommend that countries conduct a baseline assessment to determine how close they are to the proposed societal enabler targets to inform the level of investment and scale of societal enabling programmes needed to achieve them.

## Discussion

The scoping review, technical consultation and participatory process provided strong evidence that repressive legal environments, gender inequality, HIV-related stigma and discrimination, limited access to justice, and violence are impeding the global response to HIV and that societal enabling approaches to remove these impediments could have a significant impact on HIV outcomes such as HIV incidence and viral suppression. Informed by this process, the 3 S's framework, the three top-line and 15 detailed evidence-based targets, and the 15 indicators for assessing progress towards these targets, will support countries to refine program priorities, track progress, and measure the programme- and cost-effectiveness of societal enabling

**Table 2. Societal enabler targets for achievement by 2025 in the HIV sector and recommended indicators to assess progress.**

| Top-line Targets | Detailed Targets | Recommended Indicators | Baseline values based on latest Global AIDS Monitoring data and/or published study data |
|---|---|---|---|
| **Societies with supportive legal environment and access to justice** | | 1.1.1 Percentage of countries that criminalize sex work | 32.7% (36 of 110 countries) [a,b] |
| 1. Less than 10% of countries have legal environments that impede HIV services | 1.1 <10% of countries criminalize sex work, possession of small amounts of drugs, same-sex behavior and HIV transmission, exposure or non-disclosure by 2025 | 1.1.2 Percentage of countries that criminalize possession of small amounts of drugs | 76.6% (82 of 107 countries) [a] |
| | | | 38.3% (41 of 107 countries) [a,c], 49.5% (53 of 107 countries) [a,d] and 53.3% (57 of 107 countries) [a,e] |
| | | 1.1.3 Percentage of countries that criminalize same-sex sexual behavior | 35.1% (68 of 194 countries) [a] |
| | | 1.1.4 Percentage of countries that criminalize HIV transmission, exposure or non-disclosure | 60.0% (117 of 194 countries) [a] |
| | 1.2 >90% of countries have mechanisms in place for people living with HIV and key populations[b] to report abuse and discrimination and seek redress by 2025 | 1.2.1 Percentage of countries that have formal redress mechanisms in place for people living with HIV and key populations to report abuse and discrimination and seek redress | 66.2% for civil society (86 of 130 countries) [a,f] |
| | | | 68.5% for national authorities (87 of 127 countries) [a,f] |
| | | 1.2.2 Percentage of countries that have informal redress mechanisms in place for people living with HIV and key populations to report abuse and discrimination and seek redress | 66.2% for civil society (86 of 130 countries) [a,f] |
| | | | 68.5% for national authorities (87 of 127 countries) [a,f] |
| | 1.3 >90% of people living with HIV and key populations have access to legal services by 2025 | 1.3.1 Percentage of countries that have mechanisms in place for accessing affordable legal services | 89.1% for civil society (90 of 101 countries) [a] |
| | | | 96.0% for national authorities (97 of 101 countries) [a] |
| | 1.4 >90% of people living with HIV who experienced rights abuses have sought redress by 2025 | 1.4.1 Percentage of people living with HIV who have experienced rights abuses in the last 12 months and sought redress | 3.5% (27 countries) [g,h] |
| **Gender equal societies** | | In past 12 months: | |
| 2. Less than 10% of women, girls and key populations experience gender inequality and violence. | 2.1 <10% of women and girls experience IPV[a] by 2025 | 2.1.1 Percentage of women and girls subjected to IPV | 17.5% (10 countries) [a,h] |
| | 2.2 <10% of key populations[c] experience physical or sexual violence by 2025 | 2.2.1 Percentage of sex workers subjected to physical or sexual violence | 32% - 55% (any or combined workplace violence in the past year, 3 studies) [i] |
| | | | 48.4% (sex workers living with HIV experienced physical or sexual violence in past 6 months) (27 countries) [g,h] |
| | | 2.2.2 Percentage of gay men and other men who have sex with men subjected to physical or sexual violence | 11.8% - 45.1% (past year physical violence, 3 studies, US) [j] |
| | | | 7.3%-33.3% (past year sexual violence, 3 studies, US) [j] |
| | | | 54.2% (any IPV, 1 study, US) [j] |
| | | | 28.9% (MSM living with HIV experienced physical or sexual violence in past 6 months) (27 countries) [g,h] |
| | | 2.2.3 Percentage of transgender people subjected to physical or sexual violence | 16.7% (past year physical IPV, 74 studies) [k] |
| | | | 10.8% (past year sexual IPV, 74 studies) [k] |
| | | 2.2.4 Percentage of people who inject drugs subjected to physical or sexual violence | No data available. |
| | 2.3 <10% of people support inequitable gender norms by 2025 | 2.3.1 Percentage of people who support inequitable gender norms | 28.2% (11 countries, Men) [h,l,m] |
| | | | 36.6% (14 countries, Women) [h,l,m] |
| | 2.4 >90% of HIV services are gender-responsive by 2025 | 2.4.1 Percentage of HIV prevention, care and treatment services that are responsive to the differing needs of clients based on gender | No data available |

*(Continued)*

**Table 2.** (Continued)

| Top-line Targets | Detailed Targets | Recommended Indicators | Baseline values based on latest Global AIDS Monitoring data and/or published study data |
|---|---|---|---|
| **Society free of stigma and discrimination** | | In past 12 months: | 7.8% (27 countries) [g,h] |
| 3. Less than 10% of people living with HIV and key populations experience stigma and discrimination. | 3.1 <10% of people living with HIV report internalised stigma by 2025 | 3.1.1 Percentage of people living with HIV who report internalised stigma | 21.5% (Zambia and South Africa) [n] |
| | 3.2 <10% of people living with HIV report experienced stigma and discrimination in healthcare and community settings by 2025 | 3.2.1 Percentage of people living with HIV who report experienced stigma and discrimination in healthcare settings | 7.5% (Zambia and South Africa) [n] |
| | | 3.2.2 Percentage of people living with HIV who report experienced stigma and discrimination in community settings | 17.6% (27 countries) [g,h] |
| | | | 25.7% (Zambia and South Africa) [n] |
| | 3.3 <10% of key populations report experienced stigma and discrimination by 2025 | 3.3.1 Percentage of sex workers who report experienced stigma and discrimination | No data available |
| | | 3.3.2 Percentage of gay men and other men who have sex with men who report experienced stigma and discrimination | No data available |
| | | 3.3.3 Percentage of transgender people who report experienced stigma and discrimination | No data available |
| | | 3.3.4 Percentage of people who inject drugs who report experienced stigma and discrimination | No data available |
| | | 3.3.5 Percentage of sex workers who report avoiding health care because of stigma and discrimination | 7.5% [h,o] (21 countries) |
| | | 3.3.6 Percentage of gay men and other men who have sex with men who report avoiding health care because of stigma and discrimination | 10.4% [h,o] (19 countries) |
| | | 3.3.7 Percentage of transgender people who report avoiding health care because of stigma and discrimination | 6.3% [h,o] (5 countries) |
| | | 3.3.8 Percentage of people who inject drugs who report avoiding health care because of stigma and discrimination | 27.0% [h,o] (8 countries) |
| | 3.4 <10% of general population reports discriminatory attitudes towards people living with HIV | 3.4.1 Percentage of population who report discriminatory attitudes towards people living with HIV | 56.6% [h,j,p] (20 countries) |
| | | | 66.4% [h,j,q] (13 countries) |
| | 3.5 <10% of health workers report negative attitudes towards people living with HIV by 2025 | 3.5.1 Percentage of health workers who report negative attitudes towards people living with HIV | Agree that PLHIV should feel ashamed of themselves<br>• Mean: 35.3% (Bangladesh) [r]<br>• Mean: 15.7% (range: 5.3–54.7%) (China, Dominica, Egypt, Kenya, Puerto Rico, St. Christopher & Nevis) [s] |
| | | | Agree that people get infected with HIV because they engage in immoral/irresponsible behaviors<br>• 58.0% (Bangladesh) [r]<br>• 29.6% (Zambia) [t]<br>• 26.2% (South Africa) [t] |
| | 3.6 <10% of health workers report negative attitudes towards key populations by 2025 | 3.6.1 Percentage of health workers who report negative attitudes towards sex worker | Agree they prefer not to provide services to sex workers<br>• 5.3% (Bangladesh) [r]<br>• 8.0% (Zambia) [t]<br>• 9.4% (South Africa) [t] |
| | | | Agree they "put me at higher risk" of acquiring disease<br>• 19.7% (1 Bangladesh) [r] |
| | | | Agree they engage in immoral/irresponsible behavior<br>• 51.0% (1 Bangladesh) [r]<br>• 82.0% (Zambia) [t]<br>• 59.1% (South Africa) [t] |
| | | 3.6.2 Percentage of health workers who report negative attitudes towards gay men and other men who have sex with men | Agree they prefer not to provide services to men who have sex with men<br>• 14.3% (Bangladesh) [r]<br>• 10.9% (Zambia) [t]<br>• 8.9% (South Africa) [t] |
| | | | Agree they "put me at higher risk" of acquiring disease<br>• 20.7% (Bangladesh) [r] |
| | | | Agree they engage in immoral behavior<br>• 49.3% (Bangladesh) [r]<br>• 78.3% (Zambia) [t]<br>• 48.0% (South Africa) [t] |
| | | 3.6.3 Percentage of health workers who report negative attitudes towards transgender people | Agree they prefer not to provide services to transgender people<br>• 5.7% (Bangladesh) [r] |
| | | | Agree they "put me at higher risk" of acquiring disease<br>• 16.7% (Bangladesh) [r] |
| | | | Agree they engage in immoral/irresponsible behavior<br>• 39.3% (Bangladesh) [r] |
| | | 3.6.4 Percentage of health workers who report negative attitudes towards people who inject drugs | No data available |

(Continued)

**Table 2.** (Continued)

| Top-line Targets | Detailed Targets | Recommended Indicators | Baseline values based on latest Global AIDS Monitoring data and/or published study data |
|---|---|---|---|
| | 3.7 <10% of law enforcement officers report negative attitudes towards key populations by 2025 | 3.7.1 Percentage of law enforcement officers who report negative attitudes towards sex workers | No data available |
| | | 3.7.2 Percentage of law enforcement officers who report negative attitudes towards gay men and other men who have sex with men | No data available |
| | | 3.7.3 Percentage of law enforcement officers who report negative attitudes towards transgender people | No data available |
| | | 3.7.4 Percentage of law enforcement officers who report negative attitudes towards people who inject drugs | No data available |

[a] From NCPI

[b] selling sexual services is criminalized

[c] drug use or consumption is a specific offence in law

[d] possession of drugs for personal use is specified as a criminal offence

[e] drug use or consumption is specified as a criminal offence

[f] formal and informal mechanisms are not currently disaggregated

[g] from PLHIV Stigma Index 1.0 collected in 27 countries between 2008 and 2017 using snowball sampling

[h] median value

[i] Dearing et al. (2013) A Systematic Review of the Correlates of Violence Against Sex Workers

[j] Finneran et al. (2013) Intimate Partner Violence among Men Who Have Sex with Men: A Systematic Review

[k] Peitzmeier et al. (2020) Intimate Partner Violence in Transgender Populations: Systematic Review and Meta-analysis of Prevalence and Correlates

[l] from Demographic and Health Surveys (DHS)

[m] composite indicator for men and women who agreed with any one of the reasons for wife beating (all ages)

[n] Jones et al. (2020) The association between HIV-stigma and antiretroviral therapy adherence among adults living with HIV: Baseline findings from the cohort study of the HPTN 071 (PopART) trial in Zambia and South Africa

[o] from Global AIDS Monitoring (GAM) data

[p] discriminatory practices: would not purchase vegetables from a person living with HIV

[q] discriminatory practices (composite): would not purchase vegetables from a person living with HIV and/or children living with HIV should not be allowed in schools

[r] Geibel et al. (2016) Stigma Reduction Training Improves Healthcare Provider Attitudes Toward, and Experiences of, Young Marginalized People in Bangladesh

[s] Nyblade et al. (2013) A brief, standardized tool for measuring HIV-related stigma among health facility staff: results of field testing in China, Dominica, Egypt, Kenya, Puerto Rico and St. Christopher & Nevis

[t] Krishnaratne et al. (2020) Stigma and Judgment Toward People Living with HIV and Key Population Groups Among Three Cadres of Health Workers in South Africa and Zambia: Analysis of Data from the HPTN 071 (PopART) Trial.

approaches for integration into their HIV responses. Key areas for coaction across development sectors, and linked indicators, were also identified.

Modelling data suggest that decriminalization of occupations and behaviors that place people at higher risk of HIV will be an important approach for countries to pursue [16, 19]. Greatly reducing intimate partner and sexual violence will also be critical [19], as will reducing the age of consent for HIV testing to less than 16 years of age [17]. Gender inequality continues to stand in the way of global HIV goals, increasing HIV risk and impeding access to HIV services for women, girls, gay men and other men who have sex with men, transgender people, and sex workers alike [50]. A noted gap in the evidence reviewed was the lack of data linking violence with HIV acquisition for gay men and other men who have sex with men and transgender people. Yet these populations experience high levels of gender-based violence globally [51] and are at higher risk of HIV infection–up to 22 times higher among men who have sex with men [52] and 12 times higher among transgender individuals [53]. Ensuring gender-responsive HIV services [54], scaling-up gender-transformative programmes [55] and intensifying efforts to achieve gender equality through shifting harmful gender norms and addressing violence will be critical for achieving global HIV goals [56].

Despite decades of efforts to reduce HIV and key population stigma and discrimination globally [10, 57, 58], these barriers to HIV prevention, care and treatment persist. While the scale and scope of such efforts may have been insufficient to achieve large-scale and lasting change, it is also possible that societal enabling approaches to reduce stigma and discrimination thus far have not directly targeted specific domains of stigma, or addressed legal barriers to non-discrimination, that have been linked directly to HIV outcomes. Our review demonstrated that anticipated and experienced discrimination [29] and anticipated, perceived and internalized stigma are key domains of stigma that must be addressed. While the negative influence of HIV stigma and discrimination on HIV prevention, care and treatment outcomes is well documented, only recently has evidence emerged linking internalized stigma with poorer viral suppression [31, 35, 37, 38]. While previous research has found associations between stigma related to being gay or transgender with poorer access to HIV services [59–62], more research is needed to examine the link between key population-specific stigma and other HIV outcomes to inform appropriate mitigation strategies that can address intersectional stigma [63]. It is now clear that achieving universal access to biomedical interventions alone will not be enough to reach the >90% effective prevention targets and the 95-95-95 treatment targets. Societal enabling approaches designed to mitigate specific domains of HIV and key population stigma and discrimination, alongside efforts to increase gender equality, foster supportive legal environments and ensure access to justice, will also be required.

A few limitations should be noted. First, some gaps in the evidence base made it difficult to set evidence-based targets for all aspects of each societal enabler. For example, no quantitative studies were identified on the impact of access to justice or violence experienced by key populations on HIV outcomes, although there is qualitative data to support a link between improved access to justice and improved HIV outcomes [64], as well as evidence on the influence of access to justice and violence on health outcomes more broadly [65]. The wide consultations involved in the process to re-envision the enablers of the HIV response allowed for inclusion of additional targets to capture these key societal enabling approaches [47]. While work will be needed to establish baseline values, develop or adapt measurement tools, and integrate them into routine data collection for some of the proposed indicators, the majority of indicators can be reported starting in 2022. Second, none of the studies reviewed assessed the cost or cost-effectiveness of the societal enabling approaches evaluated, which may slow adoption of these approaches at the country-level. While costing and cost-effectiveness research exists for HIV interventions and social and behaviour change programs, there is a dearth of evidence that specifically examines the cost-effectiveness of approaches that address societal enablers for HIV outcomes. Cost-effectiveness analysis compares the cost per unit outcome (e.g. new HIV diagnosis, new treatment initiation, new client virally suppressed, etc.) between two or more programmes [66]. Such data would be especially helpful given the large number of societal enabling approaches that have been piloted and found to positively influence the effectiveness of HIV services. Research is urgently needed to address this gap.

The availability of numerous, evidence-based approaches for removing societal and legal impediments to HIV services, including 63 programmes to reduce stigma and discrimination [67], 5 programmes to reduce legal barriers [67] and 36 programmes to address gender inequality in the HIV response [11], will facilitate progress towards achieving the societal enabler targets. The clear articulation in the new framework of what societal enablers are and how they can impact the HIV response will support ongoing efforts, like the Global Fund's Breaking Down Barriers Initiative [9], the Global Commission on HIV and the Law [68] and the Global Partnership for action to eliminate all forms of HIV-related stigma and discrimination [69], to ensure that we can meet the 2030 HIV goals. In addition, the proposed indicators will help identify where gaps in the response exist for which institutional actors can be held

accountable. The new targets should have a substantial impact on HIV acquisition and disease progression if implemented. They will also galvanize advocacy to increase programme effectiveness, improve mathematical modelling efforts to estimate resource needs, document impact on HIV outcomes, and inform qualitative process evaluation to help understand mechanisms of change. We urge the world to move fast towards their achievement. Removing the societal and legal impediments to HIV services is critical if we are to end the AIDS epidemic as a public health threat by 2030.

## Supporting information

**S1 Checklist. Preferred Reporting Items for Systematic reviews and Meta-Analyses extension for Scoping Reviews (PRISMA-ScR) checklist.**
(PDF)

**S1 Table. Search strategy.**
(DOCX)

**S2 Table Study and intervention characteristics, HIV outcomes assessed, and study findings on key areas for development co-action from 16 studies.**
(DOCX)

**S3 Table Societal enabler target for achievement by 2025 in the development sector and recommended indicators to assess progress.**
(DOCX)

**S1 Fig. Psuedo PRISMA flowchart.**
(TIF)

**S2 Fig. Available baseline values for proposed indicators of the legal environment.**
(TIF)

**S3 Fig. Percentage of countries with proposed gender equality and stigma and discrimination indicators below or above the recommended targets.**
(TIF)

**S1 Data.**
(XLSX)

## Acknowledgments

The authors wish to thank the members of the Technical Expert Group on Social Enablers and HIV, which include the authors of this paper and Gayane Arustamyan, Adria Cots Fernandez, Leo Beletsky, Judy Chang, Michaela Clayton, Ivan Cruickshank, Tri Do, Carlos Garcia de Leon Moreno, Jankuloski Hristijan, Lise Jamieson, Ralph Jurgens, Matthew Kavanagh, Renatta Langlais, Allan Maleche, Gesine Meyer-Rath, Ruth Morgan Thomas, Wame Mosime, Lillian Mworeko, Alessandra Nilo, Laura Nyblade, Roberto Paulo, Edna Paunde, Sam Prasad, Preecha D Prempee, Kate Shannon, John Stover, Omar Syarif, Douglas Webb, Aditia Taslim, and Leigh Ann Van Der Merwe, and the following representatives from the UNAIDS Secretariat: Andrea Boccardi, Luisa Cabal, Chris Fontaine, Luisa Frescura, Peter Ghys, Peter Godfrey-Fausset, Shannon Hader, Erik Lamontagne, Laurel Sprague, Dasha Ocheret Matyushina, Angelo Mendoza, Ruth Laibon Masha, and Anna Yakusik, for their thoughtful input throughout this process. We also wish to thank the UN co-sponsors who reviewed and provided feedback on previous versions of this manuscript. In addition, we are grateful for the input of Dr.

Lucie Cluver on previous versions of this manuscript. ALS, JAI-L and GA conceptualized the manuscript. ALS led the scoping review process, contributed to the abstract and full-text review and drafted the first version of this article. TP developed and implemented the search strategy, led the title, abstract and full-text review and data abstraction processes, and drafted sections of the paper. JAI-L, GA, TB, LF, LO, SM, JP, AI and VB provided critical review of this article.

## Author Contributions

**Conceptualization:** Anne L. Stangl, Triantafyllos Pliakas, Jose Antonio Izazola-Licea, George Ayala.

**Data curation:** Anne L. Stangl, Triantafyllos Pliakas.

**Formal analysis:** Anne L. Stangl, Triantafyllos Pliakas.

**Funding acquisition:** Anne L. Stangl.

**Investigation:** Anne L. Stangl, Triantafyllos Pliakas.

**Methodology:** Anne L. Stangl, Triantafyllos Pliakas.

**Supervision:** Anne L. Stangl.

**Visualization:** Triantafyllos Pliakas.

**Writing – original draft:** Anne L. Stangl, Triantafyllos Pliakas.

**Writing – review & editing:** Jose Antonio Izazola-Licea, George Ayala, Tara S. Beattie, Laura Ferguson, Luisa Orza, Sanyukta Mathur, Julie Pulerwitz, Alexandrina Iovita, Victoria Bendaud.

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
