## [Decision Letter · Decision Letter 0]

23 Nov 2021

PONE-D-20-39747

Removing the societal and legal impediments to the HIV response: an evidence-based framework for 2025 and beyond

PLOS ONE

Dear Dr. Stangl,

Thank you for submitting your manuscript to PLOS ONE. My sincere apologies for the lengthy delay for the review.  I had considerable difficulties securing external reviewers, which seems to be more the norm during the current COVID-19 pandemic.  After one review and an additional review from myself,  I feel that your manuscript has merit but does not fully meet PLOS ONE’s publication criteria as it currently stands. Therefore, I invite you to submit a revised version of the manuscript that addresses the points raised during the review process.  I especially agree with the reviewer's comment regarding the need for a specific, but brief, discussion delineating the differences between effectiveness and cost-effectiveness with respect to evaluation of HIV interventions.  

We look forward to receiving your revised manuscript.

Kind regards,

Nickolas D. Zaller

Academic Editor

PLOS ONE

Journal Requirements:

2. Please complete and upload a PRISMA-ScR Checklist, which can be found at http://www.prisma-statement.org/documents/PRISMA-ScR-Fillable-Checklist_11Sept2019.pdf.

"N/A"

"N/A"

Reviewers' comments:

Reviewer's Responses to Questions

**Comments to the Author**

1. Is the manuscript technically sound, and do the data support the conclusions?

Reviewer #1: Yes

2. Has the statistical analysis been performed appropriately and rigorously? 

Reviewer #1: I Don't Know

3. Have the authors made all data underlying the findings in their manuscript fully available?

Reviewer #1: Yes

4. Is the manuscript presented in an intelligible fashion and written in standard English?

Reviewer #1: Yes

5. Review Comments to the Author

Reviewer #1: 1. The article would benefit from a short discussion of the difference between effectiveness and cost-effectiveness in evaluating HIV interventions (line 110). We have long known that there is a positive causative relation between many of the interventions described and reduced HIV transmission / increased treatment access and adherence. Missing is cost-effectiveness data (e.g. unit cost of HIV infection averted) which would justify massive scale up of investment in these interventions when compared with other social interventions. The Global Fund invests less than 1% of grants in so-called ‘human rights programmes.’ This will only change when the data show the relative cost-effectiveness of the proposed programmes. We are waiting for these data from the Global Fund’s Breaking Down Barriers initiative and other sources.

2. Similarly, in the Discussion (p.29) it is stated that [this approach] ‘… will allow countries to cost and integrate programmes …’ This may only happen if these programmes are demonstrated to be cost-effective when compared with other interventions such as universal testing and treatment. Other obstacles remain, such as moral and religious objections to the removal of legal obstacles, however these too can be addressed with multiple strategies involving advocacy, education, and dialogue. It is not suggested that the paper should discuss these interventions here - only the cost-effectiveness issue, because that is directly germane to the arguments made by the authors.

3. The abstract states that ‘…no global targets exist to spur funding and action…’ As the authors will be aware, in June 2021 the UN General Assembly adopted the 2021 Political Declaration on HIV/AIDS, which contains relevant commitments and targets. The article should now be updated to reflect these developments otherwise it will be out of date and misleading.

4. Line 212: ‘Inequitable gender norms were associated with more HIV testing…’ Should this be ‘less HIV testing’? Compare with line 215 ‘Community mobilization interventions to reduce IPV led to increased HIV testing…’ If correct, this may need explanation.

6. PLOS authors have the option to publish the peer review history of their article (what does this mean?). If published, this will include your full peer review and any attached files.

Reviewer #1: No

---

## [Author Response · Author response to Decision Letter 0]

21 Jan 2022

Editor's comment: I especially agree with the reviewer's comment regarding the need for a specific, but brief, discussion delineating the differences between effectiveness and cost-effectiveness with respect to evaluation of HIV interventions. 

Response: We also agree and have added in text in response to the reviewer’s suggestion (see details below). 

Reviewer's comments 1 and 2: The article would benefit from a short discussion of the difference between effectiveness and cost-effectiveness in evaluating HIV interventions (line 110). We have long known that there is a positive causative relation between many of the interventions described and reduced HIV transmission / increased treatment access and adherence. Missing is cost-effectiveness data (e.g. unit cost of HIV infection averted) which would justify massive scale up of investment in these interventions when compared with other social interventions. The Global Fund invests less than 1% of grants in so-called ‘human rights programmes.’ This will only change when the data show the relative cost-effectiveness of the proposed programmes. We are waiting for these data from the Global Fund’s Breaking Down Barriers initiative and other sources. 2. Similarly, in the Discussion (p.29) it is stated that [this approach] ‘… will allow countries to cost and integrate programmes …’ This may only happen if these programmes are demonstrated to be cost-effective when compared with other interventions such as universal testing and treatment. Other obstacles remain, such as moral and religious objections to the removal of legal obstacles, however these too can be addressed with multiple strategies involving advocacy, education, and dialogue. It is not suggested that the paper should discuss these interventions here - only the cost-effectiveness issue, because that is directly germane to the arguments made by the authors.

Response: Thank you for this suggestion. We agree that adding a short discussion of the difference between effectiveness and cost-effectiveness in evaluating HIV intervention would improve the manuscript. We have made the following clarifications and additions in response:

On pg. 6, lines 162-164 we clarified what we mean by “optimizing the effectiveness of core HIV programmes” by adding the following phrase at the end of the sentence: 

“The process to re-envision the enablers began with an in-house review at UNAIDS (led by JAI) of current understanding of how the enablers, especially the societal enablers, optimize the effectiveness of core HIV programmes (e.g. lead to increases in uptake of HIV testing, initiation of treatment, adherence to treatment, etc.).”

With regard to cost-effectiveness, we have added the following text at the end of the first paragraph of the Discussion on pg. 28, lines 462-464: 

“Informed by this process, the 3 S’s framework, the three top-line and 15 detailed evidence-based targets, and the 15 indicators for assessing progress towards these targets, will support countries to refine program priorities, track progress, and measure the programme- and cost-effectiveness of societal enabling approaches for integration into their HIV responses”.

We have also added the following limitation on pg. 29, lines 514-522: 

“Second, none of the studies reviewed assessed the cost or cost-effectiveness of the societal enabling approaches evaluated, which may slow adoption of these approaches at the country-level. While costing and cost-effectiveness research exists for HIV interventions and social and behaviour change programs, there is a dearth of evidence that specifically examines the cost-effectiveness of approaches that address societal enablers for HIV outcomes. Cost-effectiveness analysis compares the cost per unit outcome (e.g. new HIV diagnosis, new treatment initiation, new client virally suppressed, etc.) between two or more programmes [66]. Such data would be especially helpful given the large number of societal enabling approaches that have been piloted and found to positively influence the effectiveness of HIV services. Research is urgently needed to address this gap.”

Reviewer's comment 3: The abstract states that ‘…no global targets exist to spur funding and action…’ As the authors will be aware, in June 2021 the UN General Assembly adopted the 2021 Political Declaration on HIV/AIDS, which contains relevant commitments and targets. The article should now be updated to reflect these developments otherwise it will be out of date and misleading.

Response: Thank you for noting this. We had submitted the paper prior to the high-level meeting but agree that the text should now be updated throughout to reflect the outcomes of the high-level meeting. We have done so throughout the abstract, at the end of the introduction, and in the results sub-section on proposed targets for monitoring progress on societal enablers or lifting impediments. All edits made are noted in the tracked changes version of the revised manuscript we uploaded. 

Reviewer's comment 4: Line 212: ‘Inequitable gender norms were associated with more HIV testing…’ Should this be ‘less HIV testing’? Compare with line 215 ‘Community mobilization interventions to reduce IPV led to increased HIV testing…’ If correct, this may need explanation.

Response: Thank you for noting this. We have clarified our summary of the results of this study in the results section on pg. 18-19, lines 297-311: 

“ A study in South Africa that examined the influence of inequitable gender norms on HIV service use behaviours found that both women and men living with HIV who endorsed inequitable gender norms were less likely to be currently taking antiretrovirals, (i.e., women who endorsed norms accepting men’s control over and violence towards women; men as the main / sole decision-maker in a couple; and men as reluctant to seek care/help during illness; and men who endorsed norms around men as the main/sole decision maker in a couple). This study also found that receiving an HIV test in the past year was significantly associated with endorsement of inequitable gender norms (among women only, and especially for norms suggesting women have the primary/exclusive responsibility as family caretaker). While unexpected, additional analyses conducted by the study authors suggested that the association was likely due to the greater likelihood of testing after having children/during pregnancy, as HIV testing is routinely offered at antenatal services in South Africa, and as women with children were more likely to endorse those primary caretaker norms.” We also updated the data reported in Table 1 for this study to reflect the additional data reported/summarized in the paper. See tracked changes on pg. 12.

---

## [Decision Letter · Decision Letter 1]

8 Feb 2022

Removing the societal and legal impediments to the HIV response: an evidence-based framework for 2025 and beyond

PONE-D-20-39747R1

Dear Dr. Stangl,

We’re pleased to inform you that your manuscript has been judged scientifically suitable for publication and will be formally accepted for publication once it meets all outstanding technical requirements.

Kind regards,

Nickolas D. Zaller

Academic Editor

PLOS ONE

Additional Editor Comments (optional):

Reviewers' comments:

Reviewer's Responses to Questions

**Comments to the Author**

1. If the authors have adequately addressed your comments raised in a previous round of review and you feel that this manuscript is now acceptable for publication, you may indicate that here to bypass the “Comments to the Author” section, enter your conflict of interest statement in the “Confidential to Editor” section, and submit your "Accept" recommendation.

Reviewer #1: All comments have been addressed

2. Is the manuscript technically sound, and do the data support the conclusions?

Reviewer #1: (No Response)

3. Has the statistical analysis been performed appropriately and rigorously? 

Reviewer #1: (No Response)

4. Have the authors made all data underlying the findings in their manuscript fully available?

Reviewer #1: (No Response)

5. Is the manuscript presented in an intelligible fashion and written in standard English?

Reviewer #1: (No Response)

6. Review Comments to the Author

Reviewer #1: (No Response)

7. PLOS authors have the option to publish the peer review history of their article (what does this mean?). If published, this will include your full peer review and any attached files.

Reviewer #1: No

---

## [Editor Report · Acceptance letter]

11 Feb 2022

PONE-D-20-39747R1 

Removing the societal and legal impediments to the HIV response: an evidence-based framework for 2025 and beyond 

Dear Dr. Stangl:

I'm pleased to inform you that your manuscript has been deemed suitable for publication in PLOS ONE. Congratulations! Your manuscript is now with our production department. 

Kind regards, 

on behalf of

Dr. Nickolas D. Zaller 

Academic Editor

PLOS ONE